# Fabrication and Characterization of Activated Carbon Fibers from Oil Palm Trunk

**DOI:** 10.3390/polym12122775

**Published:** 2020-11-24

**Authors:** Jian Lin, Rattana Choowang, Guangjie Zhao

**Affiliations:** Beijing Key Laboratory of Wood Science and Engineering, Beijing Forestry University, Beijing 100083, China; rchoowang@sohu.com (R.C.); zhaows@bjfu.edu.cn (G.Z.)

**Keywords:** oil palm trunk, activated carbon fibers, porous structure, chromium removal

## Abstract

To develop more valuable application, oil palm trunk was successfully converted into activated carbon fibers (ACFs). An effective process of chemical treatment with dilute sulfuric acid was conducted to improve the thermal stability of primary fibers for further heating treatment. Carbon dioxide (CO_2_) was used as activator to produce much porous structure with various pore diameter. The specific surface area (S_BET_) and total pore volume (V_total_) of resultant ACFs showed increasing trend as rise of activation temperature and time. The ACFs obtained under the temperature of 900 °C and time of 90 min exhibited highest S_BET_ and V_total_, which were more than 1800 m^2^/g and 0.7 mL/g, respectively. Meanwhile, more graphic carbon on the surface of ACFs were destroyed with prolonging activation time, resulting in the oxygen-containing functional groups formed during activation process with CO_2_. Due to the abundant pores and surface functional groups, the ACFs exhibited excellent adsorption capacity of chromium and would be an alternative material for industrial adsorption utilization.

## 1. Introduction

As one of the most significant members in carbon family, activated carbon fibers (ACFs) are fibrous carbonaceous material and generally prepared from carbon fiber or precursor fibers by activation with some oxidized gas or chemical reagent at high temperature. During activation, large amounts of well-defined porous structures were formed through reaction between activator and fibers, leading to excellent adsorption properties and various applications, such as separation, electronic materials, purification, storage of natural gas and catalysts [1]. In general, ACFs were commercially manufactured by the procedures of spinning, thermostabilization, carbonization and activation from pitch, viscose rayon phenolic resin and polyacrylonitrile [2,3,4,5]. However, such raw materials for ACFs preparation are environment-unfriendly and limited resources. Thus, more and more research forced on the biomass resources as raw materials for fabricating ACFs because of sustainability and low-cost. Wood and lignin were often considered as suitable raw materials for ACFs preparation in previous studies [6,7,8,9,10]. Various processes have been developed to convert such bio-based raw materials into ACFs with large amount of porous structure, which can be applied in several significant fields, such as energy industry and wastewater treatment as well as gas phase adsorption.

The oil palm tree is an important agricultural biomass resource. Around 21.2 million ha of oil palm plantations were established worldwide for commercial oil production [11]. However, their oil productivity generally diminished after around 25 years old growth, resulting in replant of oil palm trees. This means that the massive amount of old felled oil palm trunks was produced at oil palm plantations, and it is essentially to deal with them in time because of large volume. Various methods had been explored for utilization of oil palm trunk, such as saccharification and fermentation for bio-ethanol [12], acid hydrolysis for cellulose nanocrystals [13,14], pyrolysis for bio-oil [15], mechanical processing for composite materials including laminated wood and plywood as well as particleboard [16,17,18]. To produce high value-added materials, preparation of ACFs from oil palm trunk seems to be a promising process.

Oil palm trunk contains high amount of carbohydrate polymers including cellulose and hemicellulose as well as starch, which are easily converted into liquefaction for eco-friendly products. In our previous study, oil palm trunk had been successfully liquefied with polyethylene glycol 400 (PEG 400) [19]. Precipitation of reaction mixture was performed to obtain insoluble substrate, named ligno-humin. The substrate contained around 60% of carbon, indicating a good precursor for fabricating carbonaceous materials. Besides, ligno-humin showed high molecular mobility at elevated temperature and could be directly subjected to melt-spinning without any further modification.

Accordingly, to broaden the utilization of oil palm trunk, in this article we describe the effects directed at the fabrication and characterization of ACFs. The effective chemical treatment of primary fibers was conducted to promote the conversion from fusible fibers into infusible fibers, which can be subjected to thermal treatment with high temperature. The pore structure of resultant ACFs were characterized by nitrogen adsorption and the surface chemical properties were determined with X-ray photoelectron spectroscopy. The adsorption capacity of ACFs were evaluated via liquid phase adsorption of chromium.

## 2. Materials and Methods

### 2.1. Materials and Reagents

30-year-old oil palm trunk was harvested from a replantation in Surat Thani province, Thailand. It was pulverized into powder with 40–60 mesh and then oven-dried before use. All chemicals including PEG 400, glycerol, sulfuric acid (H_2_SO_4_), potassium dichromate (K_2_Cr_2_O_7_), sodium hydroxide (NaOH) and hydrochloric acid (HCl) were purchased from commercial reagent suppliers and used as received in this experiment.

### 2.2. ACFs Preparation

Oil palm trunk powder were added into the mixture of PEG 400 and glycerol (4 to 1, *w*/*w*) as liquefaction solvent with the weight ratio of 1 to 3 in an around bottom flask which was set up with magnetic stirrer, thermometer, and reflux condenser system. The 98% sulfuric acid used as catalysis was added in flask with 2 wt% of the liquefaction solvent. Then flask was immersed in a silicone oil bath that was preheated at 180 °C for 30 min. Afterward, the flask was cooled to room temperature in cold water bath to end the reaction. The 80% 1,4-dioxane aqueous solution was added to the reaction product and then the resultant mixture was filtrated through filter paper to separate filtrated and residue. The 1,4-dioxane in filtrate was removed by evaporation. The resulting solution was then dropped into distilled water with continuous stirring for 1 h. The precipitate was collected via filtration method and then freeze dried to obtained powder which was transformed into primary fiber by melt spinning at the temperature of 180 °C.

Subsequently, the obtained primary fiber was immersed in 10% sulfuric acid solution for 2 h at room temperature. After the specified time, the primary fiber was taken out from the acid solution and washed three times with distilled water. After dried in an oven at 60 °C for 2 h, the incubated fiber was heated in air from room temperature up to 250 °C with the heating rate of 5 °C/min and held for 1 h in an electric furnace to obtain thermostabilized fiber. The carbonization and activation of thermostabilized fiber were carried out in the modified electrical tube furnace equipped with the gas inlet system. The thermostabilized fiber was heated from the room temperature to 600 °C with a heating rate of 5 °C/min and held for 1 h, and then the temperature increased to 700 to 900 °C with an equal heating rate of 10 °C/min and was maintained at the determined temperature for 30 min to 90 min. The samples are referred to as T700-t60, T800-t60, T900-t60, T900-t30, and T900-t90, respectively. During the carbonization and activation process the flowing rate of N_2_ gas was 0.1 L/min, and the CO_2_ as activated gas was applied in the period of activation process with the flow rate of 0.3 L/min. The ACF was obtained after the temperature being reduced to room temperature under N_2_ gas.

### 2.3. Characterization

Image analysis was applied for the assessment of the fusibility of all primary fibers. The assessment of the fusibility of primary fibers was conducted with a microscope connected to a digital camera (Panasonic). The morphology in the cross-section and surface of fibers were observed by using scanning electron microscopy (SEM, Gemini SEM 500, Kanagawa, Japan) at an accelerating voltage of 15 kV. The fibers were sputtered with platinum under vacuum conditions before observation. The functional groups of each sample were determined by Fourier transform infrared (FTIR) spectrometry using the GX FT-IR system (PerkinElmer, Norwalk, CT, USA) over the scan range from 400 cm^−1^ to 4000 cm^−1^. A 1 mg sample of sample powder was mixed with 99 mg of finely ground potassium bromide and compressed to a disk for measurement. Thermogravimetric analysis (TGA) was carried out with TGA 1HT/1577 (Metter, Toledo, Germany). A sample was heated from room temperature to 600 °C with a heating rate of 10 °C/min under an argon flow rate of 50 mL/min. The nitrogen adsorption-desorption isotherms were measured by using the Autosorb IQ surface area analyses (Autosorb-iQ; Quantachrome Instruments, Boynton Beach, FL, USA) for investigating the specific surface area and pore properties of ACF. A sample was degassed in a vacuum at 300 °C for 3 h before measurement at 77 K. The specific surface area (S_BET_) was calculated using the Brunauer-Emmett-Teller (BET) model [20]. The total pore volume (V_tot_) was determined from a relative pressure (*P/Po*) of 0.99. The micropore surface area and micropore volume were determined by following the t-plot method. While the Barrett-Joyner-Halends (BJH) method was applied for determining the mesopore surface area and mesopore volume, respectively [21,22,23]. The pore size distribution was calculated using the density functional theory (DFT) method [24]. The X-ray diffractometer was used with Cu Ka radiation at wavelength of 0.154 nm under 40 kV and 30 mA. The scanning rate was 2°/min from 5° to 80°. The X-ray photoelectron spectroscopy (XPS, Thermo Fischer ESCALAB 250 xi) was applied to investigate the functional groups on the surface of ACF with a monochromated Al Ka X-ray source (hν = 1486.6 eV). Each measurement was conducted twice to make sure the repeatability.

### 2.4. Chromium Removal Experiments

The chromium (Cr(VI)) removal efficiency were conducted to examine the adsorption capacity of ACFs. The stock Cr(VI) solution with the concentration of 1000 mg/L was prepared before use, which was further diluted to the required concentration. A definite amount of ACFs were added into Cr(VI) solution (40–160 mg/L) in flask, respectively. The flasks were sealed with paraffin film and then shaken with a rotation speed of 150 rpm at 25 ± 1 °C for a given time (15–240 min). After adsorption, the solution was immediately centrifuged for solid-liquid separation. The residual concentrations of Cr(VI) in filtrate were determined using a UV-vis spectrophotometer at a wavelength of 540 nm [25]. The Cr(VI) removal efficiency was calculated according to Equation (1):Cr(VI) removal efficiency (%) = (*C_0_* − *C_t_*)/*C_0_* ∗ 100%(1)
where *C_0_* and *C_t_* are the Cr(VI) concentrations (mg/L) initially and at time t, respectively.

As a blank, the concentration of Cr(VI) solution with no ACFs adsorption on each occasion were also measured. All experiments were performed at least in triplicate and the data were calculated as mean values.

## 3. Results

### 3.1. Infusible Precursor of ACFs

Oil palm trunk was liquefied and then converted into primary fibers with the diameter of 47 ± 6.8 μm and tensile strength of 32.3 ± 9.6 MPa. The water contact angle was 78°, indicating the hydrophilic property. The In general, the primary fiber obtained by melt-spinning, such as lignin fibers and polyacrylonitrile fibers as well as pitch fibers as precursors for CFs or ACFs, have been thermostabilized to maintain a fiber form during subsequent heating treatment with high temperature [26,27,28]. Thus, the primary fibers were subjected to thermostabilization upon heating up to 250 °C at varying low heating rates under air atmosphere. Unexpectedly, the primary fibers were very sensitive to the heating rates during thermostabilization and contacted with one another fusing together as observed in Figure 1B, indicating the unsuccessful thermostabilization under the conditions used.

To obtain infusible fibers, the primary fibers were treated by immersion in a dilute solution of sulfuric acid with differing concentrations of the range of 2% to 10%. The resultant incubated fibers can resist the high temperature of 250 °C without melting by using a heating rate of 1 °C/min. Particularly, the primary fiber that was immersed in 8% sulfuric acid solution achieved a pre-oxidized heating rate of 3 °C/min, and the maximum heating rate of 5 °C/min could only be used for the primary fiber treated with 10% sulfuric acid solution. These may be attributed to the partial cleavage of PEG moiety from primary fibers after incubation with diluted acid, which can be confirmed by decreasing in the intensity of C-H stretching vibration of methylene groups at 2930 cm^−1^ as showed in Figure 2A. Besides, the above result might be also caused by the diluted sulfuric acid solution failing to act as a catalyst for poly-condensation and cross-linking with the structure of the ligno-humin based primary fibers at low temperatures, resulting in the reformation of furanics in humin into phenolics and benzofurans [29,30,31]. For the thermal degradation characterization, the obtained incubated fiber showed more stable than primary fiber (Figure 2B), indicating that the chemical treatment with a dilute sulfuric acid solution at room temperature was helpful to improve the thermal stability of the primary fiber. After pre-oxidation, the oxidized fiber had great ability to resist temperatures and showed little weight loss, which can be subjected to carbonization and activation processes for the ACFs preparation.

### 3.2. Porosity Characterization of ACFs

The primary fiber obtained by melt spinning from liquefication of oil palm trunk was converted into ACFs through series of heating treatments. The activation temperature and time are the main key factors affecting the pore properties and characterization of ACFs. The effect of different activation temperatures in the rage of 700 °C to 900 °C and activation time from 30 min to 90 min on the porous structure of ACFs were investigated. The nitrogen adsorption-desorption isotherms of ACFs were given in Figure 3A. It is clearly that the ACFs prepared under different conditions of activation temperatures and times absorbed extremely high levels of nitrogen at low relative pressure (*P/P_0_*) and increased gradually as the relative pressure rose. According to the IUPAC classification, the adsorption behaviors belong to a Type I isotherm, indicating the microporous material [32]. However, at a high relative pressure above 0.4 the occurrences of some mesopore was also represented, especially at 900 °C. Besides, the nitrogen adsorption ability of each of the ACFs raised when the activation temperature increased, the same as the amount of S_BET_ and V_tot_ (as Table 1). The average pore dimension increased from 1.99 to 2.19 nm when the temperature increased from 700 °C to 900 °C. The maximum peak of pore size distribution was indicated to be around 1.0 nm (Figure 3B). The activation temperature at 900 °C showed the highest S_BET_ and V_tot_. Therefore, it was chosen for further study on the effect of activation time from 30 min to 90 min by fixing the CO_2_.

The activation time had obviously affected on the development of the S_BET_ and pore properties of ACFs. At 90 min it was 1865 m^2^/g which was contented with the highest S_BET_ of micropore of 1132 m^2^/g, corresponding to the nitrogen adsorption-desorption isotherm (Figure 3A). It also pointed out to micropore material mixed with some amount of mesopore. Remarkably, a prolonged holding time caused to an increase of mesopore. While all pore volumes increased as a function of time increased, the percentage of micropore dropped at 90 min. The mesopore ratio was increased gradually from 35.8 to 59.3% after the time was changed from 30 min to 90 min in the same way as the average pore dimension which was shown largest at 90 min of 2.27 nm. However, the pore size distribution of all activating time indicated that the shape peak between 0.75 to 1.4 nm, which is the pore size of micropore (Figure 3B). Therefore, the use of a high activation temperature and long period of activation time significantly influenced the increased reaction between activating gas and carbon atom on the surface of precursor fiber which promoted to the creation of wide pores and had a trend to break due to the merging of some micropores into mesopores [33].

### 3.3. Morphology

Figure 4 illustrates the surface morphology of ACFs as a function of the activation temperatures and times. The results obtained from SEM analysis can be seen to confirm that the CO_2_ activating gas has a good ability for removing the carbon atoms from the surface of precursor fiber. It can clearly be seen that more various sizes of pore appeared on the fiber surface, compared to the original precursor fibers which have smooth surfaces. In consideration of the effect of activation temperature at a constant of the activation time for 60 min and CO_2_ flow rate at 300 mL/min, the ACFs activated at 700 °C showed a lot of tiny pinholes and some have wide pore on their surface. The pinhole pores were subsequently enlarged with the increase of the activation temperature, meanwhile, the new pores were also developed. Especially at 900 °C, the resulting ACFs indicated the rising ratio of enlarged pores, corresponding with the pore properties that are shown in Table 1. The dominant pores in the resulting ACFs were micropore, however, the portion of mesopores increased when a high activation temperature and prolonged activation time were applied. Obviously, all the surfaces of the resulting ACFs activated at 900 °C and the various times from 30 min to 90 min by fixing the CO_2_ flow rate at 300 mL/min appeared to have enlarged pores.

### 3.4. Surface Chemistry

Generally, the XPS is widely used to determine the changes in chemical bonding states and concentrations of surface functional groups that have high influences on the adsorption capacity of porous carbon material as well as the porosity properties [34]. The broad scan of XPS spectra and atomic concentrations on the surface of ACFs activated with different times at 900 °C and precursor fibers as a reference were presented in Figure 5A. All samples showed the similar XPS spectra patterns and indicated the main peaks in the position of C1s and O1s around 290 to 282 eV and from 548 to 528 eV, respectively, suggesting that carbon atom and oxygen atom were the major component elements for ACFs. Compared with precursor fibers, ACFs showed relative low oxygen atom content. This may be caused by the de-oxygenation and aromatization of the structure of fibers under heating condition especially at high temperature, and surface oxides were removed by using CO_2_ as an oxidizing gas during activation process [35]. With the increase of activation time, the carbon atom content increased while the oxygen atom content decreased, revealing that more oxygen atom suffered content loss as prolonging the activating time. This was also a process of carbon atom enrichment.

The curve fitted high resolution XPS scans of T900-t90 for C1s spectrum were presented in Figure 5B as an example. The C1s spectra was fitted into five individual component peaks [36]. Table 2 summarized the results of the fits of C 1s region for all ACFs. The values in% intensity for graphitic carbon and oxygen-containing groups showed obvious differences among ACFs. A decreasing trend could be observed in graphitic carbon, whereas an opposite trend for carbon bonded to oxygen containing functions as the activation time increasing, indicating the higher extent of activation by CO_2_. Besides, the carbonyl groups and carboxylic groups as well as hydroxyl groups showed obvious increase, while the rest groups (CO_3_^2−^, CO, CO_2_) were showed the opposite changed tendency. Thus, it is believed that more graphitic carbons reacted with CO_2_ molecules to generate more functional groups containing C-O, which may be attributed to be an advantage on the chemical adsorption.

### 3.5. Cr(VI) Removal Capacities

Cr(VI) is one kind of the most important constituents among toxic compounds in the effluents, which may enter into body and cause serious health problems. Adsorption treatment is a fast and effective technology for the eradication of Cr(VI). Thus, all the ACFs were subjected to adsorb the Cr(VI) from aqueous solutions and each maximum adsorption capacity of ACFs were determined as shown in Figure 6. As increasing the activation time, the adsorption capacity of ACFs gradually increased. Especially, the highest adsorption capacity of 39.67 mg/g for T900-t90 was obtained, which was near two times greater than that of T900-t30. These results can be attributed to the porous structure and oxygen-containing functional groups on the surface of ACFs. Besides, the Cr(VI) adsorption value achieved in this study was higher than that of cotton-based ACFs and also comparable to that of AC and its fibric composite [37,38,39]. Consequently, ACFs in this study could be considered as an alternative material for the Cr(VI) removal.

## 4. Conclusions

Oil palm trunk was liquefied with PEG-400 and then successfully converted into ACFs with porous structure by carbonization and CO_2_ activation. The chemical treatment with dilute sulfuric acid improved the thermal stability of primary fibers for following heating processes with high temperature. More pores, especially for the mesopores, were formed during activation process via increasing activating temperature and time. The ACFs prepared with T900-t90 in this study exhibited the highest S_BET_ of around 1800 m^2^/g and V_tot_ of around 0.7 mL/g. Besides, the amount of oxygen-containing functional groups showed the same trend with S_BET_ and pore volume with increasing activation time, which is attributed to high adsorption capacity of Cr(VI). Therefore, the resultant ACFs with highly porous structure and excellent Cr(VI) adsorption would be considered as an appropriate adsorbent for industrial sorption processes.

## Figures and Tables

**Figure 1 polymers-12-02775-f001:**
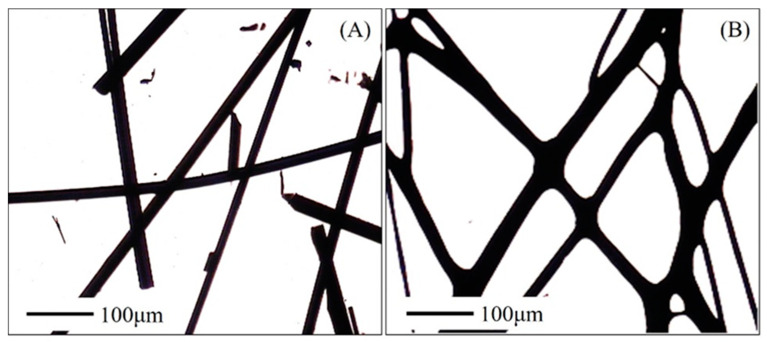
Microscopic images of the primary fiber (**A**) and thermostabilized fibers (**B**) heated in air from room temperature to 250 °C at the heating rate of 1 °C/min.

**Figure 2 polymers-12-02775-f002:**
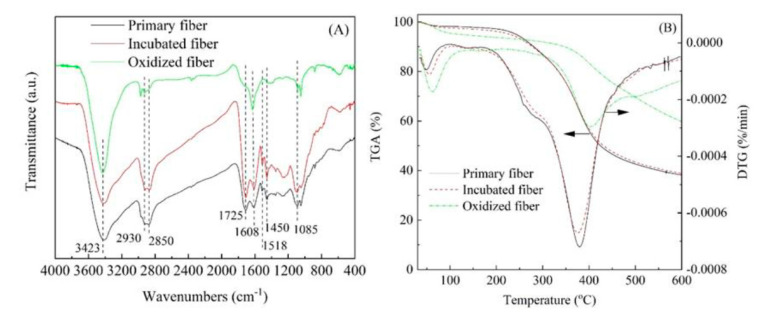
The FTIR spectrums (**A**) and TGA curves (**B**) of primary fiber, incubated fiber (10% acid solutions) and the oxidized fiber with the heating rate of 5 °C/min.

**Figure 3 polymers-12-02775-f003:**
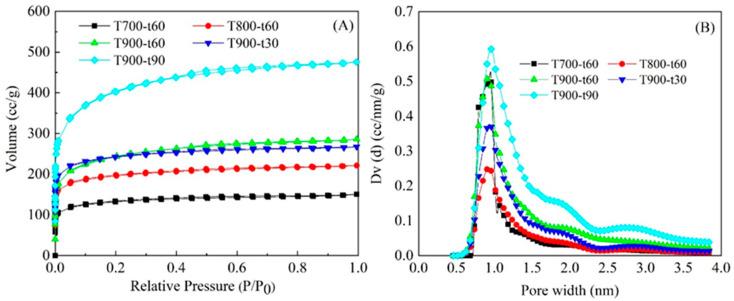
N_2_ adsorption-desorption isotherms (**A**) and pore width (**B**) of ACFs activated with different activation temperatures and times by fixing the CO_2_ gas flow rate at 300 mL/min.

**Figure 4 polymers-12-02775-f004:**
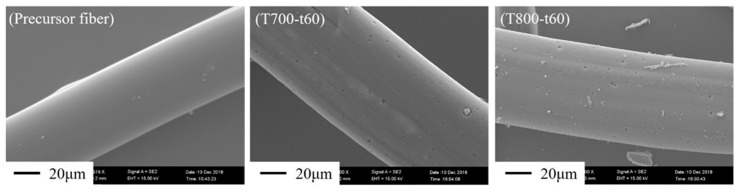
SEM micrographs of ACFs as a function of activation temperature and time.

**Figure 5 polymers-12-02775-f005:**
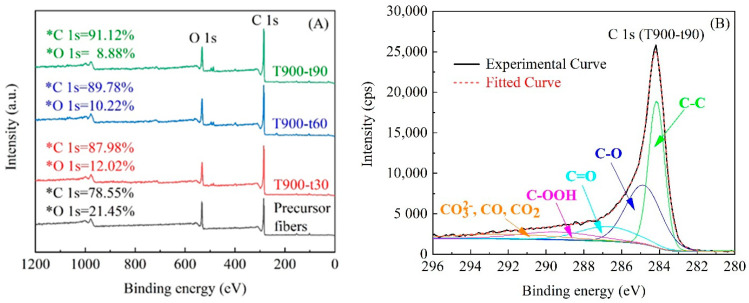
XPS spectrum of ACFs (**A**) and curve fitted high resolution XPS scans of T900-t90 for C1s (**B**).

**Figure 6 polymers-12-02775-f006:**
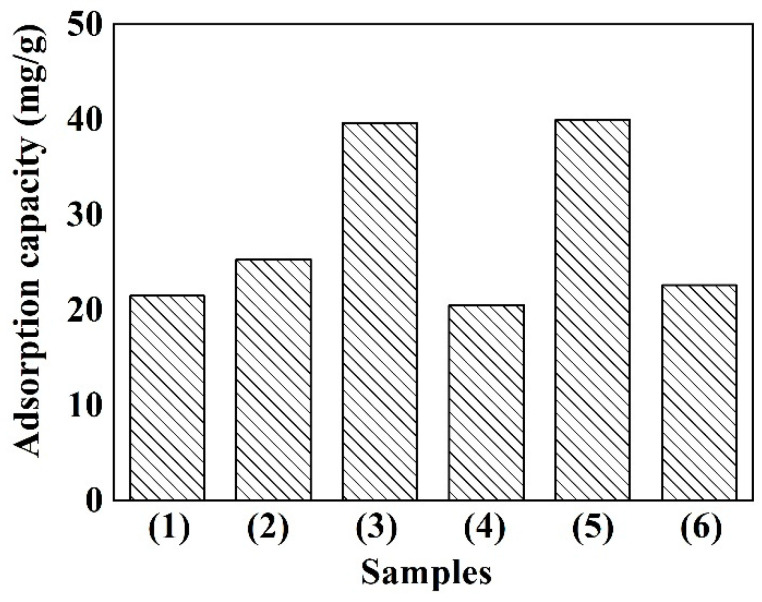
Cr(VI) adsorption capacity of various carbon materials. (1) T900-t30; (2) T900-t60; (3) T900-t90; (4) Cotton-based ACFs; (5) Bast-based AC; (6) AC/glass fiber fabric.

**Table 1 polymers-12-02775-t001:** The pore properties of ACFs.

Samples	Specific Surface Area (m^2^/g)			Pore Volume (cc/g)			%V_meso_	Average Pore Dimension (nm)
	S_BET_	S_micro_	S_meso_	V_tot_	V_micro_	V_meso_		
Precursor fibers	4.3	1.9	2.9	0.001	0.000	0.001	100	4.00
T700-t60	542	438	171	0.234	0.200	0.096	41.0	1.99
T800-t60	857	665	231	0.343	0.296	0.128	37.3	1.90
T900-t60	1018	715	435	0.442	0.366	0.234	52.9	2.19
T900-t30	1109	830	280	0.413	0.364	0.148	35.8	1.87
T900-t90	1865	1132	835	0.735	0.588	0.436	59.3	2.27

**Table 2 polymers-12-02775-t002:** Results of the fits of the C 1s region.

Samples	Peak from C 1s Spectrum Binding Energy (%)				
	C-C	C-OH	C=O	COOH	CO_3_^2−^, CO, CO_2_
T900-t30	65.9	13.6	3.7	2.9	13.9
T900-t60	58.1	20.2	4.7	4.3	12.7
T900-t90	43.5	28.8	9.2	7.9	10.6

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
