# Peer review of "Fabrication and Characterization of Activated Carbon Fibers from Oil Palm Trunk"

_polymers, 2020, doi:10.3390/polym12122775_

Round 1

Reviewer 1 Report

Review Report

Manuscript number:1009055; Title: Fabrication and Characterization of Activated Carbon Fibers from Oil Palm Trunk.

This manuscript entitled " Fabrication and Characterization of Activated Carbon Fibers from Oil Palm Trunk" by Lin et al., is a  research article, which focuses on the preparation of activated carbon fibers (ACFs) form oil palm trunk and their characterization and adsorption of chromium. However, some points need to be addressed and revised considerably before acceptance.

Comments:

  1. It would be nice if the author(s) would describe the latest developments of bio-based materials that are converted to activated carbon fibers and their potential application in the Introduction section.
  2. Authors should provide more experiments on chromium adsorption and kinetics. It would be ideal to provide a table listing the various activated carbon fibers or materials prepared from bio-based sources, for the adsorption of chromium, and compare their current work.
  3. General comment: the manuscript has language and typo errors.
  4. Fiber characteristics (primary fiber and thermostabilized fibers) such as fiber diameter, water contact angle, and mechanical properties would be specified.
  5. The adsorption mechanism of precursor fibers (T900-t90) towards the adsorption of Cr (VI) should be illustrated.
  6. Reusability and regeneration of precursor fibers (T900-t90) for the various cycles of operation for the adsorption of chromium should be provided.
  7. Author(s) should be considered the amount of total chromium, Cr (III) and Cr (VI) during the interaction between fibers and the metal ion (chromium) in the adsorption process.

Author Response

Dear Reviewer,

Thank you very much for your effort to read the manuscript and providing specific comments on how to improve the paper’s quality.

The following details were improved following your suggestions. Please find our comments next to your remarks.

1. It would be nice if the author(s) would describe the latest developments of bio-based materials that are converted to activated carbon fibers and their potential application in the Introduction section. 

Answer: This is a good suggestion. Activated carbon fibers from bio-based materials such as wood and lignin were briefly described in the first paragraph of introduction section.

2. Authors should provide more experiments on chromium adsorption and kinetics. It would be ideal to provide a table listing the various activated carbon fibers or materials prepared from bio-based sources, for the adsorption of chromium, and compare their current work.

Answer: This research is focused on the fabrication and characterization of ACFs from oil palm. For chromium adsorption of ACFs, it was conducted to check the adsorption capacity whether is good or not. The following step would pay more attention to the experiments on its detail adsorption behavior and kinetics and so on. Besides, comparation of adsorption of chromium between current work and previous work was presented in Figure 6, please check.

3. General comment: the manuscript has language and typo errors.

Answer: Thanks. We are really very sorry for the English and typo problems. We are unthoughtful in our previous manuscript. We have read and thoroughly corrected our manuscript to make the paper readable.

4. Fiber characteristics (primary fiber and thermostabilized fibers) such as fiber diameter, water contact angle, and mechanical properties would be specified.

Answer: The basic fiber information mentioned have been added in the beginning of the results section.

5. The adsorption mechanism of precursor fibers (T900-t90) towards the adsorption of Cr (VI) should be illustrated. Reusability and regeneration of precursor fibers (T900-t90) for the various cycles of operation for the adsorption of chromium should be provided. Author(s) should be considered the amount of total chromium, Cr (III) and Cr (VI) during the interaction between fibers and the metal ion (chromium) in the adsorption process.

Answer: These are very important suggestion for improving the Cr(VI) adsorption determination. The adsorption mechanism is very important for understanding the adsorption processes. The more detailed adsorption behavior will be further studied.

If there are any problems, please feel free to contact us.

We are looking forward to hearing good news.

Thanks million.

Kind regards

Jian Lin, Rattana Choowang, Guangjie Zhao

Reviewer 2 Report

The document describes the preparation of activated carbon from oil palm trunks. The idea of using a biomass material to obtain a material that can be used to decontaminate water from heavy metals, can be a sustainable option for an agroindustrial waste. The authors include a characterization of the material, using different analytical methodologies.

There are few suggestions for the document to improve results presentation:

Please, include in methodology the number of replicates done for each analysis.
Where it can be included, please present a variation of data on the graphics (i.e. standard deviation bars on the graphics).

For metal absorption analysis by the selected material, please include a table or a figure with the experiments' quantitative data. This will be of particular importance to assess the efficiency of the material prepared.

Author Response

Dear Reviewer,

Thank you very much for your effort to read the manuscript and providing specific comments on how to improve the paper’s quality.

The following details were improved following your suggestions. Please find our comments next to your remarks.

1. Please, include in methodology the number of replicates done for each analysis.

Answer: The number of replicates done for each analysis has been illustrated in characterization section.

2. Where it can be included, please present a variation of data on the graphics (i.e. standard deviation bars on the graphics).

Answer: The graphics in manuscript were normally drawn based on the data obtained from measuring instruments. Generally, there are no bars presenting in figures such as morphology and curves results.

3. For metal absorption analysis by the selected material, please include a table or a figure with the experiments' quantitative data. This will be of particular importance to assess the efficiency of the material prepared.

Answer: Thanks for good suggestion. The adsorption results in current work as well as the similar materials in previous word were presented in Figure 6, please check.

If there are any problems, please feel free to contact us.

We are looking forward to hearing good news.

Thanks million.

Kind regards

Jian Lin, Rattana Choowang, Guangjie Zhao

Round 2

Reviewer 1 Report

The revised manuscript can be accepted for publication